# Loss of prion protein control of glucose metabolism promotes neurodegeneration in model of prion diseases

Hélène Arnould[1,2], Vincent Baudouin[1,2], Anne Baudry[1,2], Luiz W. Ribeiro[1,2], Hector Ardila-Osorio[1,2], Mathéa Pietri[1,2], Cédric Caradeuc[3,4], Cynthia Soultawi[1,2], Declan Williams[5], Marjorie Alvarez[1,2], Carole Crozet[6], Fatima Djouadi[7,8], Mireille Laforge[1,2], Gildas Bertho[3,4], Odile Kellermann[1,2], Jean-Marie Launay[9,10], Gerold Schmitt-Ulms[5], Benoit Schneider[1,2]*

1 INSERM, UMR-S 1124, Paris, France, 2 Université de Paris, UMR-S 1124, Paris, France, 3 CNRS, UMR 8601, Paris, France, 4 Université de Paris, UMR 8601, Paris, France, 5 University of Toronto, Tanz Centre for Research in Neurodegenerative Diseases, Canada, 6 IRMB, Université de Montpellier, INSERM, CHU de Montpellier, Montpellier, France, 7 INSERM, UMR-S 1138, Paris, France, 8 Université de Paris, UMR-S 1138, Paris, France, 9 Assistance Publique des Hôpitaux de Paris, INSERM UMR942, Hôpital Lariboisière, Paris, France, 10 Pharma Research Department, Hoffmann La Roche Ltd, Basel, Switzerland

* benoit.schneider@parisdescartes.fr

**Data Availability Statement:** All relevant data are within the manuscript and its Supporting Information files.

## Abstract

Corruption of cellular prion protein (PrP$^C$) function(s) at the plasma membrane of neurons is at the root of prion diseases, such as Creutzfeldt-Jakob disease and its variant in humans, and Bovine Spongiform Encephalopathies, better known as mad cow disease, in cattle. The roles exerted by PrP$^C$, however, remain poorly elucidated. With the perspective to grasp the molecular pathways of neurodegeneration occurring in prion diseases, and to identify therapeutic targets, achieving a better understanding of PrP$^C$ roles is a priority. Based on global approaches that compare the proteome and metabolome of the PrP$^C$ expressing 1C11 neuronal stem cell line to those of PrP$^{null}$-1C11 cells stably repressed for PrP$^C$ expression, we here unravel that PrP$^C$ contributes to the regulation of the energetic metabolism by orienting cells towards mitochondrial oxidative degradation of glucose. Through its coupling to cAMP/protein kinase A signaling, PrP$^C$ tones down the expression of the pyruvate dehydrogenase kinase 4 (PDK4). Such an event favors the transfer of pyruvate into mitochondria and its conversion into acetyl-CoA by the pyruvate dehydrogenase complex and, thereby, limits fatty acids β-oxidation and subsequent onset of oxidative stress conditions. The corruption of PrP$^C$ metabolic role by pathogenic prions PrP$^{Sc}$ causes in the mouse hippocampus an imbalance between glucose oxidative degradation and fatty acids β-oxidation in a PDK4-dependent manner. The inhibition of PDK4 extends the survival of prion-infected mice, supporting that PrP$^{Sc}$-induced deregulation of PDK4 activity and subsequent metabolic derangements contribute to prion diseases. Our study posits PDK4 as a potential therapeutic target to fight against prion diseases.

**Funding:** This work was funded by INSERM and a project grant (TargetingPDK1inAD, n°ANR-16-CE16-0021-01) from the French Agence Nationale de la Recherche (ANR) awarded to BS. VB is funded by the ANR PrPC&PDK1 European project (n°ANR-14-JPCD-0003-01) awarded to BS and the French Association pour la Recherche sur la SLA (ARSLA foundation - n°J19D08DOC026) awarded to VB. The funders had no role in study design, data collection and analysis, decision to publish, or preparation of the manuscript.

**Competing interests:** I have read the journal's policy and the authors of this manuscript have the following competing interests: J.M.L has non-financial competing interests with Hoffmann La Roche Ltd laboratories. He acts as an expert witness for Hoffmann La Roche Ltd laboratories.

## Author summary

Transmissible Spongiform Encephalopathies (TSEs), commonly named prion diseases, are caused by pathogenic prions PrP$^{Sc}$ that trigger degeneration of neurons in the brain. Although PrP$^{Sc}$ exerts its neurotoxicity by corrupting the function(s) of normal cellular prion protein (PrP$^C$), our understanding of the mechanisms involved in prion diseases remains limited. There is still to date no medicine to fight against TSEs. The current study demonstrates that the deregulation of PrP$^C$ regulatory function towards glucose metabolism contributes to neurodegeneration in prion diseases. In the brain of prion-infected mice, PrP$^{Sc}$-induced overactivation of pyruvate dehydrogenase kinase 4 (PDK4) and downstream reduction in mitochondria pyruvate dehydrogenase (PDH) activity promote a metabolic shift from glucose oxidative degradation to pro-oxidant fatty acids β-oxidation contributing to prion pathogenesis. The pharmacological inhibition of PDK4 extends the lifespan of prion-infected mice by rescuing normal glucose metabolism. This study opens up new avenues to design PDK4-based therapeutic strategies to combat TSEs.

## Introduction

The cellular prion protein PrP$^C$ is mainly known for its implication in Transmissible Spongiform Encephalopathies (TSEs), commonly named prion diseases [1], and has been more recently involved in two other unrelated amyloid-based neurodegenerative pathologies, the Alzheimer's [2,3] and Parkinson's [4,5] diseases. In TSEs, it is well established that pathogenic prions (PrP$^{Sc}$) exert their toxicity by interacting with PrP$^C$ and corrupting its function(s) through a loss- and/or a gain-of-PrP$^C$ function [6], depending on the stage of the disease. In Alzheimer's or Parkinson's diseases, dysregulation of PrP$^C$ function(s) upon binding of oligomers of Aβ peptides [2,7] or pathological fibrils of α-synuclein [8] to common PrP$^C$ epitopes provokes synaptic failure at the root of synaptic plasticity or long term potentiation alterations. PrP$^C$ emerges as a global sensor of numerous protein conformers enriched with a β-sheet structure [9] and corruption of PrP$^C$ function(s) seems at play in several amyloid-based neurodegenerative diseases. Thus, identifying the role(s) exerted by PrP$^C$ in neurons is a prerequisite to grasp the nature and sequence of events occurring in those neurodegenerative disorders, for which several neuronal abnormalities have already been reported, including oxidative stress [10], endoplasmic reticulum stress [11], autophagy deficits [12,13], mitochondria dysfunctions [10,14] and/or energetic metabolic dysregulation [15–20].

Unfortunately, *Prnp* knockout mice (PrP$^{-/-}$) did not allow to reveal any obvious function for PrP$^C$ as PrP$^{-/-}$ mice develop normally and only display minor neurophysiological abnormalities [21–24]. The more recently derived PrP$^{null}$ ZH3 mouse line, where the *Prnp* gene was excised in the C57Bl/6J mouse genome using a TALEN approach, only indicated a possible role of PrP$^C$ in the maintenance of the interactions between neurons and Schwann cells in peripheral nerves [25]. The absence of a clear phenotype for PrP$^{null}$ mice suggested mechanisms occurring early during embryogenesis that would compensate for the lack of PrP$^C$ [26], thus making it difficult to unravel PrP$^C$ function(s) with PrP$^{null}$ mice.

From a molecular point of view, PrP$^C$ is a ubiquitous sialoglycoprotein anchored at the outer leaflet of the plasma membrane via a glycosylphosphatidylinositol moiety [27]. With the help of the 1C11 neuronal stem cell line that can convert into serotonergic or noradrenergic neurons [28], we were able to firstly assign a signaling function to neuronal PrP$^C$. In lipid rafts of the plasma membrane, PrP$^C$ plays the role of a receptor or co-receptor [29], and/or acts as a scaffolding protein that adjusts stoichiometric interactions between PrP$^C$ partners, and fine-

tunes multiple signaling pathways [27]. The identification of several PrP$^C$ interacting proteins and signaling effectors governed by PrP$^C$ allowed proposing multiple roles for PrP$^C$ dealing with neuronal differentiation [30–32], cell adhesion [33], cell survival [34,35], regulation of redox equilibrium [36], control of neurotransmitter-associated functions [37], protection against oxidative stress [38], TNFα-mediated inflammatory injury [39], and excitotoxicity through regulation of NMDA receptor nitrosylation [40]. Very recent proteomic studies comparing the interactomes of PrP$^C$ in distinct mouse cell lines revealed that the neural cell adhesion molecule 1 (NCAM1) would be the only protein that robustly interacts with PrP$^C$ across cell models tested [41,42]. Together with prior data, which established that PrP$^C$ controls NCAM1 polysialylation [43], these results underscored a critical role of PrP$^C$ in processes supported by NCAM1, including epithelial-to-mesenchymal transition, focal adhesion dynamics, neurite outgrowth and synaptic plasticity [44–47].

To further dissect and enlarge the cellular functions governed by PrP$^C$, we here conducted a comparative mass spectrometry analysis of the proteomes of 1C11 neuronal stem cells and 1C11 cells chronically silenced for PrP$^C$ expression through a shRNA-based strategy (referred to as PrP$^{null}$-1C11 cells). PrP$^{null}$-1C11 cells exhibit less than 5% of residual PrP$^C$ expression compared to parental 1C11 cells [31]. We provide evidence that PrP$^{null}$-1C11 cells displayed deep modifications in the expression level of enzymes involved in glucose metabolism with an up-regulation of the glycolysis pathway and a down-regulation of mitochondrial enzymes. NMR- and Seahorse-based functional analyses further revealed metabolic abnormalities in the absence of PrP$^C$ with a global reduction of the glycolysis flux and reduced consumption of $O_2$ by the mitochondria respiratory chain. Such metabolic abnormalities notably relate to reduced activity of the pyruvate dehydrogenase (PDH) complex that links glycolysis to mitochondria tricarboxylic acid (TCA) cycle. We show that loss of PrP$^C$ coupling to the cAMP/protein kinase A (PKA) signaling provokes a strong rise of expression of the pyruvate dehydrogenase kinase isoform 4 (PDK4), which in turn phosphorylates the PDH enzyme E1 (PDHA1), at the root of the reduced PDH activity. Importantly, the decrease of glucose oxidative degradation in the absence of PrP$^C$ leads to a metabolic switch towards fatty acids use that is further associated with the onset of oxidative stress conditions in PrP$^{null}$-1C11 cells. Finally, we provide evidence for a loss-of-PrP$^C$ metabolic role in the hippocampus of prion-infected mice with a reduction in PDH activity and a lower glucose oxidative degradation rate in favor of an increased fatty acids β-oxidation, and show the inhibition of PDK4 extends the survival of prion-infected mice. Altogether, these data support a new role of PrP$^C$ in the control of the cell energetic metabolism and the cell redox status, whose deregulation by pathogenic prions contributes to neurodegeneration, and introduce PDK4 as a potential therapeutic target to fight against prion diseases.

## Results

### The loss of PrP$^C$ affects the expression of glycolytic and mitochondrial enzymes in 1C11 cells

To identify novel functions for PrP$^C$, we performed a global analysis that compared the proteomes of PrP$^{null}$-1C11 cells [31] and their PrP$^C$-expressing 1C11 counterparts by tandem mass spectrometry MS/MS [48]. The proteome data sets collected revealed that 721 proteins were differentially expressed between PrP$^{null}$-1C11 and 1C11 cells, among which 403 proteins were down-regulated (fold changes < 0.84) and 318 were up-regulated (fold changes > 1.25) (**S1 Table**).

A Kyoto Encyclopedia of Genes and Genomes (KEGG) pathway analysis was next performed using the Database for Annotation, Visualization, and Integrated Discovery (DAVID)

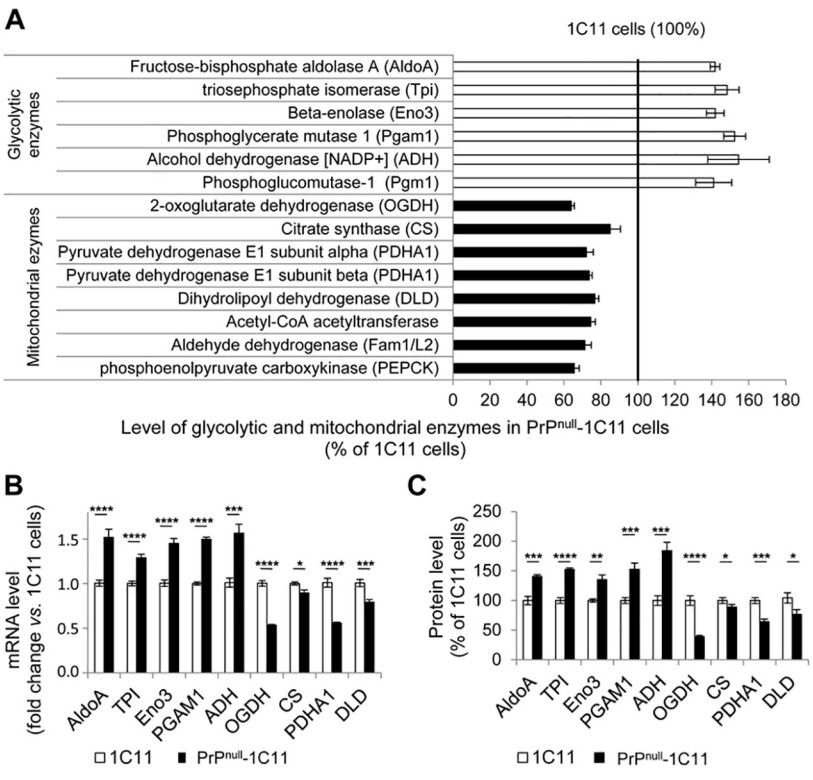

**Fig 1. Loss of PrP^C in the 1C11 neuronal cell line impacts on the expression of glycolytic and mitochondrial enzymes.** (**A**) Relative amounts of glycolytic and mitochondrial enzymes involved in the oxidative degradation of glucose in PrP^null-1C11 cells as compared to parental 1C11 cells. (**B**) Relative mRNA levels of fructose-bisphosphate aldolase A (AldoA), triosephosphate isomerase (TPI), beta-enolase 3 (Eno3), phosphoglycerate mutase 1 (PGAM1), alcohol dehydrogenase (ADH), 2-oxoglutarate dehydrogenase (OGDH), citrate synthase (CS), pyruvate dehydrogenase E1 (PDHA1) and dihydrolipoyl dehydrogenase (DLD) in PrP^null- vs. 1C11 cells (n = 10). (**C**) Histogram quantifications for AldoA, Eno3, PGAM1, TPI, ADH, OGDH, CS, PDHA1 and DLD protein expression in 1C11 and PrP^null-1C11 cells. α-tubulin was used for normalization (n = 7). Values are mean fold change ± SEM. * denotes $p < 0.05$, ** $p < 0.01$, *** $p < 0.001$ and **** $p < 0.0001$.

software to identify cellular functions affected by PrP^C depletion. This approach indicated that the structure and dynamics of the actin cytoskeleton was one of the cell functions mostly modified in PrP^null-1C11 cells (**S1A Fig**), consistent with our previous demonstration of a regulatory role of PrP^C towards actin turnover necessary for the sprouting of neurites and the neuronal polarity [31]. In addition, the absence of PrP^C deeply impacted cell metabolism. Metabolisms of amino-acids, fatty acids, and glucose (glycolysis, neoglucogenesis, Krebs cycle) were affected in PrP^null-1C11 cells (**S1A and S1B Fig**). We found in PrP^null-1C11 cells overexpression of glycolytic and fermentation enzymes, and underexpression of enzymes involved in mitochondria oxidative metabolism (**Fig 1A**). Notably, the steady-state level of the pyruvate dehydrogenase (PDH) complex was affected. PDH is a central player in energetic metabolism, that converts the glycolysis end-product pyruvate into mitochondrial acetyl-CoA and thereby self-sustains the degradative flux of glucose in cells. The expression levels of the pyruvate dehydrogenase enzyme E1 (PDHA1) and the dihydrolipoyl dehydrogenase enzyme E3 (DLD) of the PDH complex were reduced by 20–25% in PrP^null- vs. 1C11 cells (**Fig 1A**).

Nine glycolysis and mitochondrial enzymes identified by MS/MS were then selected based on low and high peptide counts, sequence coverage and fold change, both down-regulated as well as up-regulated ones, to analyze their changes of expression at the mRNA level and to validate the proteomic approach. Compared to PrP^C-expressing 1C11 cells, RT-qPCR

experiments showed increased gene transcription of the glycolytic enzymes fructose-bisphosphate aldolase A (AldoA), triosephosphate isomerase (TPI), beta-enolase 3 (Eno3), and phosphoglycerate mutase-1 (PGAM1), and the fermentative enzyme alcohol dehydrogenase (ADH) in PrP$^{null}$-1C11 cells (**Fig 1B**). As concerns the mitochondrial enzymes 2-oxoglutarate dehydrogenase (OGDH), citrate synthase (CS), PDHA1, and DLD, significant decreases in their mRNA levels were measured in PrP$^{null}$-1C11 cells *vs.* their PrP$^C$-expressing counterparts (**Fig 1B**). At the protein level, western blot experiments showed an up-regulation of AldoA, TPI, Eno3, PGAM1, and ADH, and a down-regulation of OGDH, CS, PDHA1, and DLD in PrP$^{null}$-1C11 cells, in agreement with the proteomic analysis (**Figs 1C and S2**). We concluded that in cells depleted for PrP$^C$, the alterations in the expression level of glycolytic, fermentation and mitochondrial enzymes depend on transcriptional mechanisms.

As a whole, these data support the view that PrP$^C$ may contribute to the regulation of glucose oxidative metabolism by adjusting the expression of glycolytic and mitochondrial enzymes.

## NMR-based metabolomic analysis indicates reduced glycolytic flux in PrP$^{null}$-cells

We next assessed whether an imbalance in the expression of glycolytic and mitochondrial enzymes in PrP$^{null}$-1C11 cells alters glucose metabolism. We first conducted an NMR-based analysis of metabolites present in the culture medium of both 1C11 and PrP$^{null}$-1C11 cells. A multivariate analysis and a supervised Orthogonal Projection to Latent Structures Discriminant Analysis (OPLS-DA) [49] were performed to maximize class discrimination. The OPLS-DA model showed a clear separation between PrP$^{null}$- and 1C11 cells groups according to the predictive axis t[1] (**Fig 2A**). Culture medium concentrations of lactate and glucose were the most significantly affected by the absence of PrP$^C$ (**Fig 2B**). Compared to PrP$^C$-expressing cells, a higher concentration of glucose (25%, **Fig 2B left**) and a lower concentration of the fermentation end-product lactate (20%, **Fig 2B right**) were measured in the medium of PrP$^{null}$-1C11 cells cultured for four days in the presence of 25 mM glucose. The difference in lactate concentrations between PrP$^{null}$- and PrP$^C$-expressing 1C11 cells was confirmed by measuring the extracellular acidification rates (ECAR) by the Seahorse XF$^e$96 Extracellular Flux Analyzer. In PrP$^{null}$-1C11 cells grown in 25 mM glucose, the ECAR ($17.06 \pm 0.97$ mpH/min) was significantly lower compared to the ECAR measured with parental PrP$^C$ expressing 1C11 cells ($20.84 \pm 0.70$ mpH/min) (**Fig 2C**).

The higher concentration of glucose associated with a lower level of lactate in the culture medium of PrP$^{null}$- *vs.* 1C11-cells suggested reduced glycolytic flux in cells silenced for PrP$^C$ expression. Thus, we measured the ECAR after cell treatment with oligomycin (10 mg ml$^{-1}$), which inhibits mitochondrial ATP production. This allowed to estimate the maximal glycolysis rate and the glycolytic reserve. We monitored maximal glycolysis, glycolytic reserve and found that they were both significantly lower in PrP$^{null}$-1C11 cells (35% and 30%, respectively, **Fig 2D**) than in 1C11 cells. No significant variation in glucose transporter 3 (GLUT3) protein level was observed between PrP$^{null}$- and PrP$^C$-expressing cells (**Fig 2E**). We thus excluded that the source of glycolytic impairment in PrP$^{null}$-1C11 cells originated from modification in the expression level of the rate-limiting GLUT3, the main glucose transporter present in the brain, specifically expressed by neurons (for review, see [50] and references therein).

Despite the overexpression of glycolytic enzymes in the absence of PrP$^C$, our data indicate a decreased conversion of glucose into lactate by the glycolytic/fermentation pathway, *i.e.*, a reduced glycolytic flux in PrP$^{null}$-1C11 cells, that does not depend on a reduced glucose entry in cells.

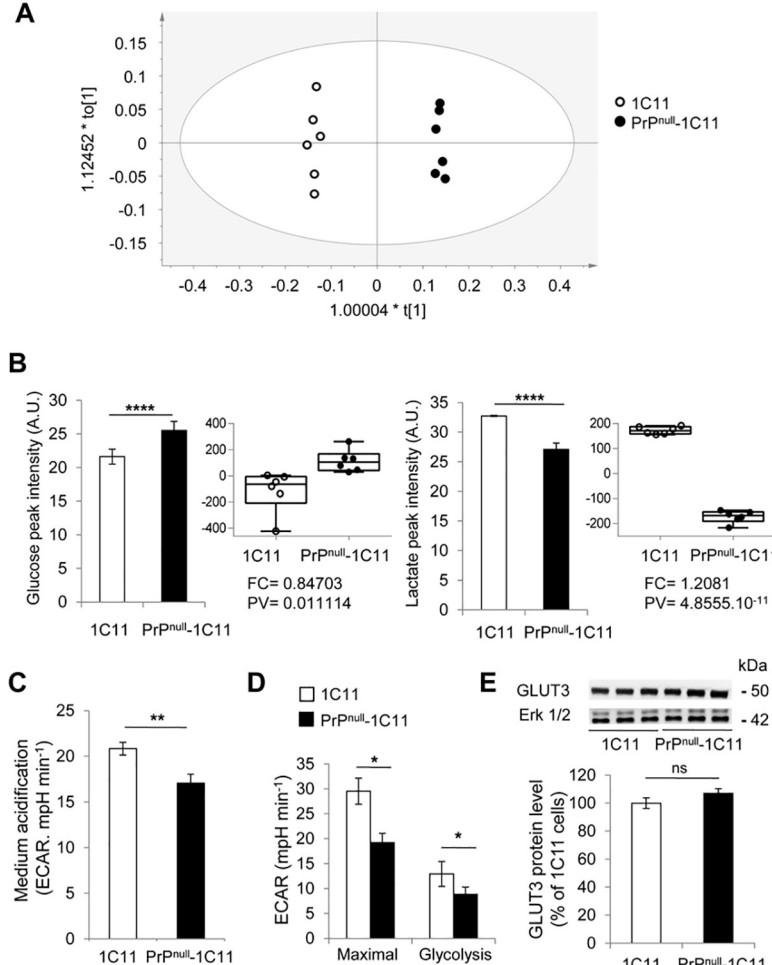

**Fig 2. Loss of PrP^C causes reduction of the glycolytic flux.** (**A**) The NMR analysis of metabolites present in the cell culture medium of PrP^null- and 1C11 cells (n = 6) grown for four days in DMEM-Glutamax-FCS with 25 mM glucose. The OPLS-DA model shows that the PrP^null-1C11 cells group is distinct from that of 1C11 cells according to the predictive axis t[1]. The orthogonal axis to t[1] is the uncorrelated variables axis. A p-value (CV-ANOVA) = 9.82218 $10^{-8}$, $R^2X$(cum) = 0.872, $R^2Y$(cum) = 0.996, and $Q^2$(cum) = 0.994 reflect the good quality of the model. (**B**) NMR-derived histograms and boxplots of external concentrations of glucose and lactate in the cell culture medium of PrP^null- and 1C11 cells (n = 6) grown for four days in DMEM-Glutamax-FCS with 25 mM glucose. Buckets at 4.12 ppm and 3.2 ppm correspond to lactate and glucose, respectively. Histograms show glucose and lactate peak intensities (in arbitrary units—A.U.). Boxplots show normalized concentrations of glucose and lactate. The fold-change (FC) and p-values (PV) are indicated on each boxplot. (**C**) Extracellular medium acidification rate (ECAR) in 1C11 and PrP^null-1C11 cells measured by Seahorse XF^e96 Extracellular Flux Analyzer. Seahorse experiments were run with the assay medium containing 25 mM glucose and Glutamax 1X. Data represent the average of 6 replicates ± SEM. (**D**) Maximal glycolysis (ECAR after inhibition of mitochondrial ATP synthase activity with oligomycin 10 mg ml$^{-1}$) and glycolytic reserve (= maximal ECAR—basal ECAR) in PrP^null- vs. 1C11 cells grown as in C (n = 10). (**E**) Western blot (cropped image) and histogram quantification for GLUT3 protein expression in PrP^null- vs. 1C11 cells. ERK1/2 MAP kinases were used for normalization (n = 7). Data are mean fold change ± SEM. * denotes p < 0.05, ** p < 0.01, *** p < 0.001, and **** p < 0.0001.

## PrP^C silencing reduces basal mitochondrial oxygen consumption without affecting the respiratory chain activity

To next explore whether the reduction of glycolytic flux in PrP^null-1C11 cells relates to some impairment of the mitochondrial function, we first measured real-time consumption of oxygen via the Seahorse XF^e96 Analyzer. The mitochondrial respiration was assessed using

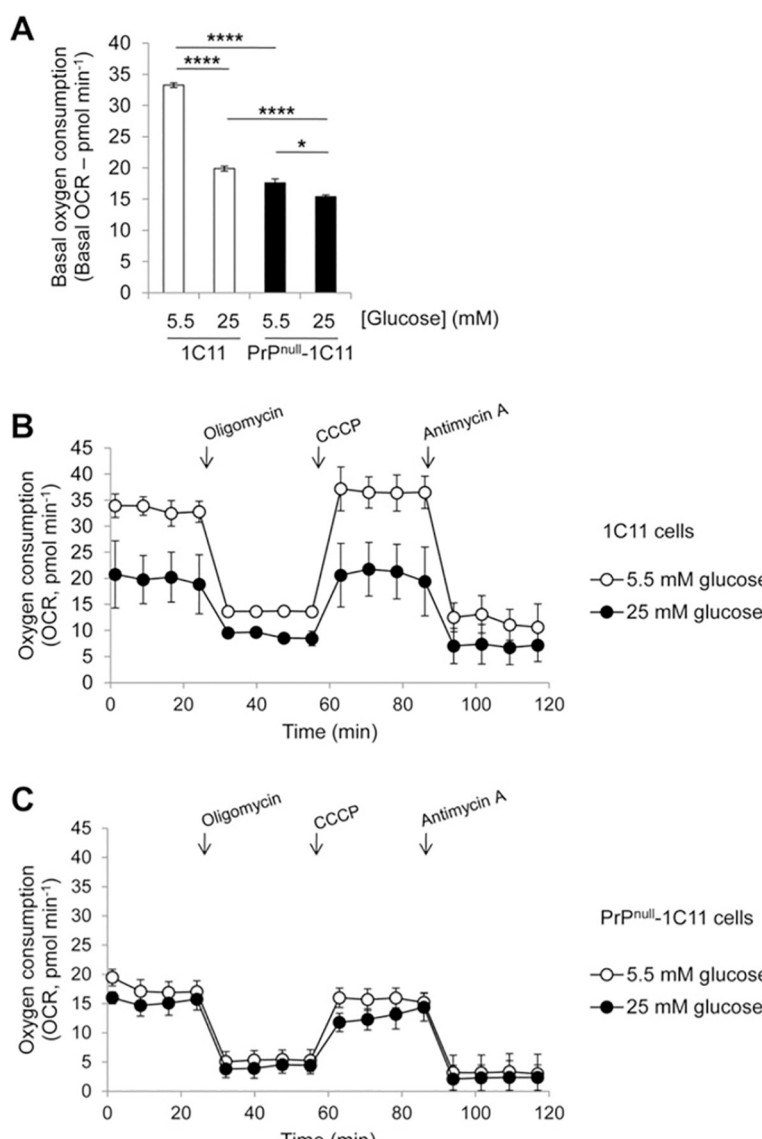

**Fig 3. PrP<sup>C</sup> depletion in 1C11 cells reduces mitochondrial oxygen consumption.** (A) Comparison of basal oxygen consumption rate (OCR) between 1C11 and PrP<sup>null</sup>-1C11 cells grown in DMEM containing 5.5 or 25 mM glucose using a Seahorse XF<sup>e</sup>96 Extracellular Flux Analyzer (n = 6). Values are the mean fold change ± SEM. (**B, C**) Mitochondrial respiration profile of 1C11 cells (**B**) and PrP<sup>null</sup>-1C11 cells (**C**) grown in a medium containing 5.5 or 25 mM glucose. The OCR was measured before and after the sequential addition of oligomycin (10 mg ml$^{-1}$), CCCP (5 μM), and antimycin A (10 μM) for 120 min. Values are the average of 6 replicates ± SEM. * denotes $p < 0.05$ and **** $p < 0.0001$.

glucose at 25 mM and 5.5 mM, as the energy substrate. The basal oxygen consumption rate (OCR) measured in PrP<sup>null</sup>-1C11 cells in the presence of 25 mM glucose was ~25% lower than that measured with 1C11 cells (**Fig 3A**). With PrP<sup>C</sup>-expressing 1C11 cells, changing glucose concentration from 25 mM to 5.5 mM in the cell culture medium enhanced basal OCR by ~70% to compensate for the decrease in glucose concentration (**Fig 3A**). Contrasting with the response of 1C11 cells, a very slight augmentation of basal OCR in PrP<sup>null</sup>-1C11 cells was shown when lowering external glucose concentration, leading to a ~50% difference in basal OCR values between 1C11 and PrP<sup>null</sup>-1C11 cells grown with 5.5 mM glucose (**Fig 3A**). These

results indicated that PrP$^C$ depletion attenuates the basal mitochondria respiration rate and renders mitochondria insensitive to variations of external glucose concentration.

To next characterize the mitochondrial respiration profile, the OCR was measured during 120 min after the addition in the culture medium of (i) oligomycin (10 mg ml$^{-1}$), the inhibitor of ATP synthase, (ii) CCCP (5 μM), the uncoupling agent, and (iii) antimycin A (10 μM), the inhibitor of the mitochondrial complex III of the respiratory chain. **Fig 3B** depicts the time-dependent changes in OCR in PrP$^C$-expressing 1C11 cells. Oligomycin treatment significantly reduced basal OCR independently of glucose concentration. This was followed by a rapid increase of 10% over basal OCR when CCCP was injected. Mitochondria respiration was significantly inhibited upon the injection of antimycin A, as indicated by an OCR decrease of 65% compared to the basal OCR (**Fig 3B**). With PrP$^{null}$-1C11 cells, the changes in OCR induced by oligomycin, CCCP, and antimycin A were comparable to those observed with 1C11 cells (**Fig 3C**).

We concluded that while PrP$^C$ depletion modifies basal oxygen consumption by mitochondria, the absence of PrP$^C$ does not change the mitochondria respiratory chain activity and its chemo-osmotic coupling to the ATP synthase.

## The loss of PrP$^C$ provokes a deficit of PDH complex activity

In PrP$^{null}$-1C11 cells, the incapacity of mitochondria to adjust the OCR to variations of external glucose concentration supports the view of some deficit in the entry of pyruvate in the mitochondria and conversion into acetyl-CoA through the PDH complex. The proteomic approach already indicated reduced expressions of PDHA1 (44 ± 1%) and DLD (21 ± 3%) in PrP$^{null}$-1C11 cells (**Fig 1A**). In addition, we further measured a 42 ± 7% decrease in the total enzymic activity of PDH in PrP$^{null}$-1C11 cells (**Fig 4A**). Beyond the reduced protein expression of PDH subunits, the decrease in PDH activity also resulted from a 1.3-fold increase of the phosphorylation state of PDHA1 on Ser293 (**Fig 4B**) that was reported to lower PDH activity [51].

These data suggested that PrP$^C$ positively controls the activity of the PDH complex. Such a PrP$^C$ control permits to efficiently fuel pyruvate into mitochondria, and thereby to optimize glucose consumption and the rate of the glycolytic flux.

## The deficit of PDH activity in PrP$^{null}$-cells and primary PrP$^{0/0}$ hippocampal neurons depends on PDK4 over-expression

The pyruvate dehydrogenase kinases (PDKs) are well-known negative regulators of PDH activity. PDKs phosphorylate PDHA1 and thereby block the entry of pyruvate in the mitochondria and its conversion into acetyl-CoA [52]. Among all four PDK subtypes, RT-qPCR analyses revealed a selective, strong increase of PDK4 mRNA (~50-fold) in PrP$^{null}$-1C11 compared to 1C11 cells (**Figs 4C and S3**). Such a huge augmentation of PDK4 transcripts was followed by an enhanced PDK4 expression at the protein level (~25-fold) (**Fig 4D**). The increase in PDK4 level in PrP$^{null}$-1C11 cells promoted the rise in PDHA1 phosphorylation and the reduction in PDH activity, as the inhibition of PDK4 with dichloroacetate (DCA, 2 mM) in PrP$^{null}$-1C11 cells strongly reduced PDHA1 phosphorylation (**Fig 4B**) and increased PDH activity by 25% (**Fig 4A**).

Such a PrP$^C$-dependent control of the PDK4-PDH axis was not restricted to the 1C11 neuronal cell line. Transient siRNA-based PrP$^C$ silencing in PC12 neuronal cells also provoked a rise in PDK4 protein expression level (~2.5-fold) as compared to PC12 cells transfected with an unrelated siRNA (**S4A Fig**). As for PrP$^{null}$-1C11 cells, the decrease in PDH activity

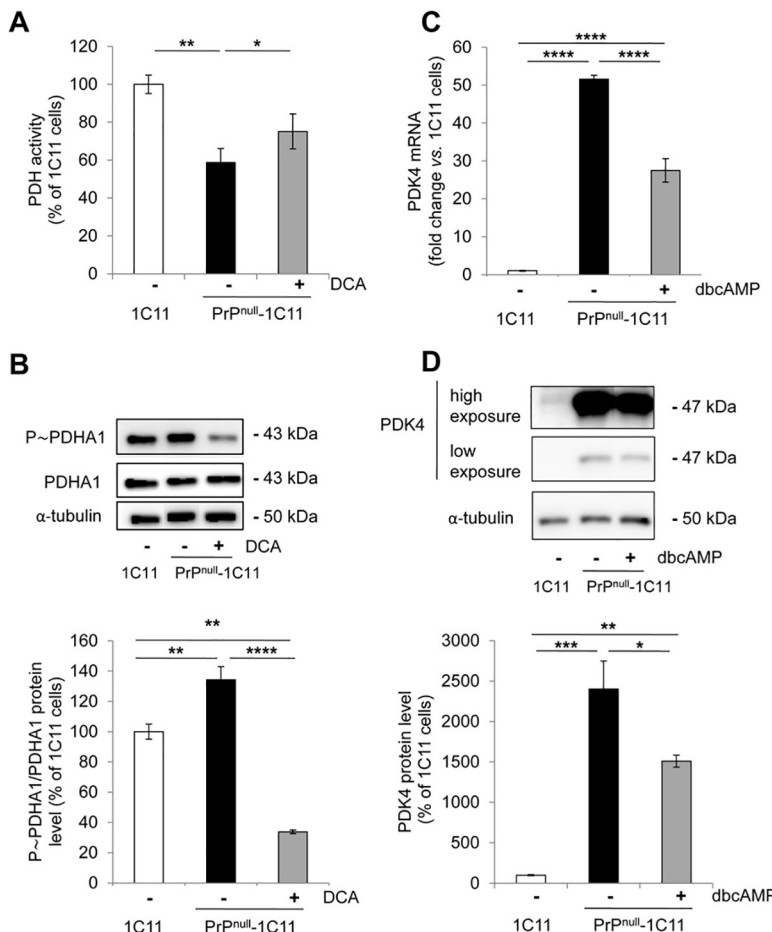

**Fig 4. PDK4 overexpression in PrP^null-1C11 cells is at the root of the reduction of PDH activity.** (**A**) PDH activity in 1C11 cells and PrP^null-1C11 cells exposed or not to DCA (2 mM, 6h) (n = 5). (**B**) Representative western blot and histogram quantifications of the P~PDHA1 over PDHA1 ratio in cells treated or not with the PDK4 inhibitor DCA (2 mM, 6h) (n = 6). α-tubulin was used for normalization. (**C**) Relative mRNA levels of PDK4 between 1C11, PrP^null-1C11, and PrP^null-1C11 cells treated for 24 h with dbcAMP (1 mM) as assessed by RT-qPCR (n = 10). (**D**) Representative western blot and histogram quantifications of PDK4 protein level in 1C11 and PrP^null-1C11 cells treated or not with dbcAMP (n = 7). α-tubulin was used for normalization. Data are the mean ± SEM. * denotes $p < 0.05$, ** $p < 0.01$ and **** $p < 0.0001$.

(48 ± 2%) measured in PC12 neuronal cells silenced for PrP^C expression was partly counteracted upon PDK4 inhibition with DCA (2 mM, 6 h) (**S4B Fig**).

Finally, to get closer to the *in vivo* situation, we established primary cultures of hippocampal neurons isolated from 3-month-old PrP^0/0-FVB mice and their PrP^+/+-FVB counterparts. Compared to PrP^+/+-hippocampal neurons, we measured an increased PDK4 protein level (~15-fold) and a reduced PDH activity (25 ± 1%) in PrP^0/0-neurons (**S4C Fig**). The inhibition of PDK4 with DCA (2 mM, 6 h) in PrP^0/0-hippocampal neurons rescued, at least partly, PDH activity to a level that corresponded to ~88% of the PDH activity measured in PrP^+/+-hippocampal neurons (**S4D Fig**).

Our data obtained with the 1C11 and PC12 cell lines and corroborated with primary hippocampal neurons from FVB mice showed that PrP^C depletion leads to the rise in PDK4 level and the subsequent impairment of PDH activity, supporting the conclusion that PrP^C is a cell positive regulator of glucose metabolism by exerting a negative control on PDK4 expression.

## PDK4 overexpression in PrP$^{null}$-1C11 cells relates to loss of PrP$^C$ coupling to cAMP/PKA signaling

Nuclear peroxisome proliferator-activated receptor PPARγ is one of the main transcription factors contributing to the expression of several genes involved in energetic metabolism, including PDK4 [53]. Correlating with the increase in PDK4 level, we found that PPARγ transcripts were ~12-fold up-regulated (**Fig 5A**) and PPARγ protein level was ~35-fold increased (**Fig 5B**) in PrP$^{null}$- *vs*. 1C11 cells.

Interestingly, cyclic AMP (cAMP) and cAMP-dependent protein kinase A (PKA)-connected signaling pathways were shown to be involved in the downregulation of expression of both PPARγ in hepatocytes [54] or preadipocytes [55] and PDK4 in human placental trophoblasts [56]. Because PrP$^C$ activates cAMP/PKA signaling [45,57], we postulated that the up-regulation of PPARγ and PDK4 in PrP$^C$-depleted cells relates to a reduced cAMP/PKA signaling. Supporting this hypothesis, PKA stimulation with dibutyryl cyclic-AMP (dbcAMP, 1 mM) for 24 h in PrP$^{null}$-1C11 cells decreased at the mRNA and protein levels the expression of PPARγ (**Fig 5A and 5B**) and PDK4 (**Fig 4C and 4D**).

Overall, these data indicate that PrP$^C$ decreases the expression of PPARγ and PDK4 through the cAMP/PKA pathway.

## PrP$^C$ silencing provokes increases in lipid handling and oxidative stress

Increased use of fatty acids (FA) through the β-oxidation pathway has also been shown to amplify PDK4 expression [53]. We next probed the status of FA metabolism as well as the expression of enzymes involved in the lipid handling in cells silenced for PrP$^C$. The degradation of the palmitic acid was then assessed in PrP$^{null}$-1C11 cells and PrP$^C$-expressing 1C11 cells. Cells were exposed to [$^3$H]-palmitate and the production of [$^3$H]$_2$O at the end of the respiratory chain was measured. As compared to 1C11 cells, a significant increase (17 ± 2.1%) of palmitate β-oxidation was measured in PrP$^{null}$-1C11 cells (**Fig 5C**), indicating a metabolic switch towards fatty acids use in the absence of PrP$^C$. Such a rise in palmitate β-oxidation was associated with an increased FA translocation into the mitochondria as the expression level of the gene encoding the neuronal carnitine palmitoyltransferase 1C (Cpt1C) [58] was ~1.8-fold enhanced in PrP$^{null}$-cells compared to PrP$^C$-expressing 1C11 cells (**Fig 5D**). Based on a ~1.5-fold increase in the acylCoA synthetase (Acsf2) mRNA level in PrP$^{null}$- *vs*. 1C11 cells (**Fig 5D**), we also concluded for an accelerated conversion of FA into acylCoA by Acsf2, fueling the degradation of the acylCoA by the β-oxidation pathway in cells depleted for PrP$^C$.

Because the β-oxidation pathway and oxidative phosphorylation in the mitochondria produce ATP that was shown to also stimulate the PDK activity (for review, see [59] and references therein), the concentration of ATP was measured in PrP$^{null}$-cells. As compared to parental cells, the ATP level was increased by ~65% in cells silenced for PrP$^C$ (**Fig 5E**).

Combined with the reduced glycolytic flux (**Fig 2**), these data show that the loss of PrP$^C$ in 1C11 cells changes in fuel preference from glucose towards fatty acids. This creates a vicious circle that maintains PDK4 activity at a high level, which in turn self-sustains down-regulation of glucose consumption and the glycolytic rate in favor of increased degradation of fatty acids.

Excessive mobilization of the mitochondria activity by the β-oxidation pathway was shown to generate high levels of reactive oxygen species (ROS) and the accumulation of β-oxidation by-products, exerting deleterious effects [60]. With the help of the CM-H$_2$DCFDA fluoroprobe that detects the superoxide anion and hydrogen peroxide, we measured an increase (50 ± 6%) in the level of ROS in PrP$^{null}$- *vs*. 1C11 cells (**Fig 5F**). Such a rise in the ROS level was accompanied by the onset of oxidative stress conditions. As compared to PrP$^C$-expressing cells, we monitored at the mRNA level lower expression (20 ± 5%) of two antioxidant enzymes,

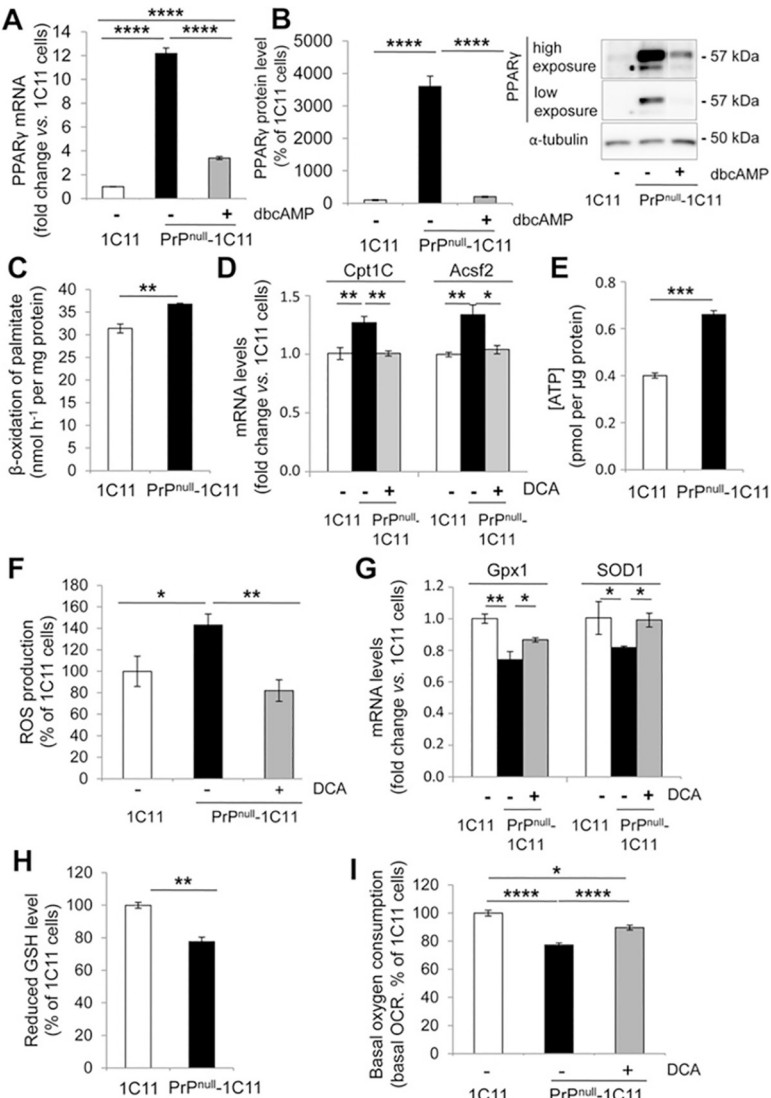

**Fig 5. PDK4-dependent PDH inhibition provokes increases of the lipid handling and oxidative stress in PrP^null^-cells.** (**A**) Relative mRNA levels of PPARγ between 1C11, PrP^null^-1C11, and PrP^null^-1C11 cells treated for 24 h with dbcAMP (1 mM), grown in a medium containing 25 mM glucose, as assessed by RT-qPCR (n = 10). (**B**) Representative western blot and histogram quantifications for PPARγ in 1C11 and PrP^null^-1C11 cells treated or not with dbcAMP. α-tubulin was used for normalization (n = 7). (**C**) Palmitate β-oxidation rate measured in 1C11 and PrP^null^-1C11 cells (n = 3). (**D**) Relative mRNA levels of Cpt1C and Acsf2 between 1C11 and PrP^null^-1C11 cells exposed or not to DCA (2 mM) for 6 h, as assessed by RT-qPCR (n = 6). (**E**) ATP levels in 1C11 and PrP^null^-1C11 cells measured by luminescence (n = 3). (**F**) ROS levels in 1C11 and PrP^null^-1C11 cells exposed or not to DCA (2 mM, 6h) measured by fluorescence (n = 6). (**G**) Relative mRNA levels of SOD1 and Gpx1 between 1C11 and PrP^null^-1C11 cells exposed or not to DCA (2 mM, 48h), as assessed by RT-qPCR (n = 3). (**H**) GSH levels in 1C11 and PrP^null^-1C11 cells (n = 6). (**I**) Basal OCR in 1C11 and PrP^null^-1C11 cells grown in 25 mM glucose in the presence or not of DCA (2 mM) for 48 h (n = 6). Data are the mean ± SEM. $^*$ denotes $p < 0.05$, $^{**}$ $p < 0.01$, $^{***}$ $p < 0.001$, and $^{****}$ $p < 0.0001$.

the superoxide dismutase 1 (SOD1) and glutathione peroxidase 1 (Gpx1) in PrP^null^-cells (**Fig 5G**). This leads to a decrease (20 ± 3%) in the level of reduced glutathione (GSH) in the absence of PrP^C^ (**Fig 5H**) as assessed using the fluorogenic Cell Tracker CMFDA.

These data indicate that PrP^C^ negative control of PDK4 expression and activity prevents a cell metabolic switch from the oxidative degradation of glucose to fatty acids β-oxidation, and thereby the onset of oxidative stress conditions.

## PDK4 inhibition cancels fatty acid-associated oxidative stress and rescues glucose oxidative degradation in PrP$^{null}$-cells

Our aim was then to assess whether the inhibition of PDK4 and the subsequent dephosphorylation of PDHA1 (**Fig 4B**) and the partial rescue of PDH activity (**Fig 4A**) reengage the metabolism towards the oxidative degradation of glucose [61] and thereby reverse the metabolic abnormalities observed in PrP$^C$-depleted 1C11 cells.

Treatment of PrP$^{null}$-1C11 cells with PDK4 inhibitor DCA (2 mM) allowed Cpt1C and Acsf2 mRNAs to return to levels highly comparable to those measured in PrP$^C$-expressing 1C11 cells (**Fig 5D**), supporting the view that DCA treatment permits to reduce fatty acids β-oxidation in PrP$^{null}$-cells. Importantly, PDK4 inhibition fully abrogated the excessive ROS production measured in PrP$^C$-depleted cells (**Fig 5F**). Moreover, the inhibition of PDK4 rescued normal antioxidant activities, as inferred by mRNA levels encoding Gpx1 and SOD1 (**Fig 5G**) comparable to those obtained in 1C11 cells. Finally, Seahorse experiments revealed that DCA treatment of PrP$^{null}$-1C11 cells significantly increased basal OCR that reached an intermediate level between PrP$^{null}$- and PrP$^C$-expressing 1C11 cells (**Fig 5I**), indicating that PDK4 inhibition restores, at least partly, glucose oxidative degradation in PrP$^C$-depleted cells.

These results indicate that PDK4 inhibition counteracts the metabolic abnormalities and the fatty acid-associated oxidative stress conditions caused by the absence of PrP$^C$.

## Prion infection down-regulates glucose oxidative degradation in mice

Corruption of PrP$^C$ functions by PrP$^{Sc}$ is at the root of prion diseases [6]. As common molecular changes have been evidenced in prion infection and PrP$^C$ loss-of-function experimental paradigms [3,39,62,63], we next assessed the status of glucose oxidative metabolism under prion infection.

C57Bl6/J mice were inoculated via the intracerebral route with cell-based inocula either uninfected (SHAM) or infected with the Fukuoka prion strain (Fk6-mice) [63]. SHAM mice remained healthy over more than 250 days post-inoculation and prion-infected mice died at 164.5 ± 2.0 days (**Fig 6A**, n = 10). Of note, the Fukuoka strain mostly displays a cortico-tropism [64], which excludes measuring any neuronal biochemical parameters in brain cortex at the end stage of the disease due to massive neuronal loss. We instead focused our metabolic analysis on the hippocampus, which is less severely infected by prions [64] and thereby allows us to grasp some PrP$^{Sc}$-induced biochemical events that occur later or with less intensity than in the cortex [63]. As compared to SHAM mice, we measured at the end-stage of the disease a ~3.5-fold increase in the protein expression level of PDK4 (**Fig 6B**) and a decrease in the activity of the PDH complex (33 ± 5%) in the hippocampus of Fk6-mice (**Fig 6C**). Supporting a reduced glycolytic flux within a prion-infectious context, we also measured a lower amount of pyruvate (25 ± 3.9%) in prion-infected mice (**Fig 6D**). Corroborating our proteomic data with PrP$^{null}$-cells, a reduced enzymatic activity (15 ± 2%) of the Krebs cycle 2-oxoglutarate dehydrogenase (OGDH) was recorded in Fk6-mice (**Fig 6E**), which further favors the view of a lower glucose oxidative degradation rate in prion-infected mice. Finally, we measured a rise in the level of acetyl-CoA (25 ± 7%) in the hippocampus of Fk6-mice *vs*. SHAM mice (**Fig 6F**). We excluded that the augmentation of acetyl-CoA in prion-infected mice originated from anaplerotic reactions that replenish α-ketoglutarate and oxaloacetate Krebs cycle intermediates, as we showed no significant variation in Glutamate Dehydrogenase (GDH) activity (**Fig 6G**) and a slight increase (6 ± 3%) in Pyruvate Carboxylase (PCx) activity (**Fig 6H**), respectively. As for PrP$^{null}$-cells, a metabolic switch towards the use of fatty acids would instead account for the increase of acetyl-CoA in Fk6-mice.

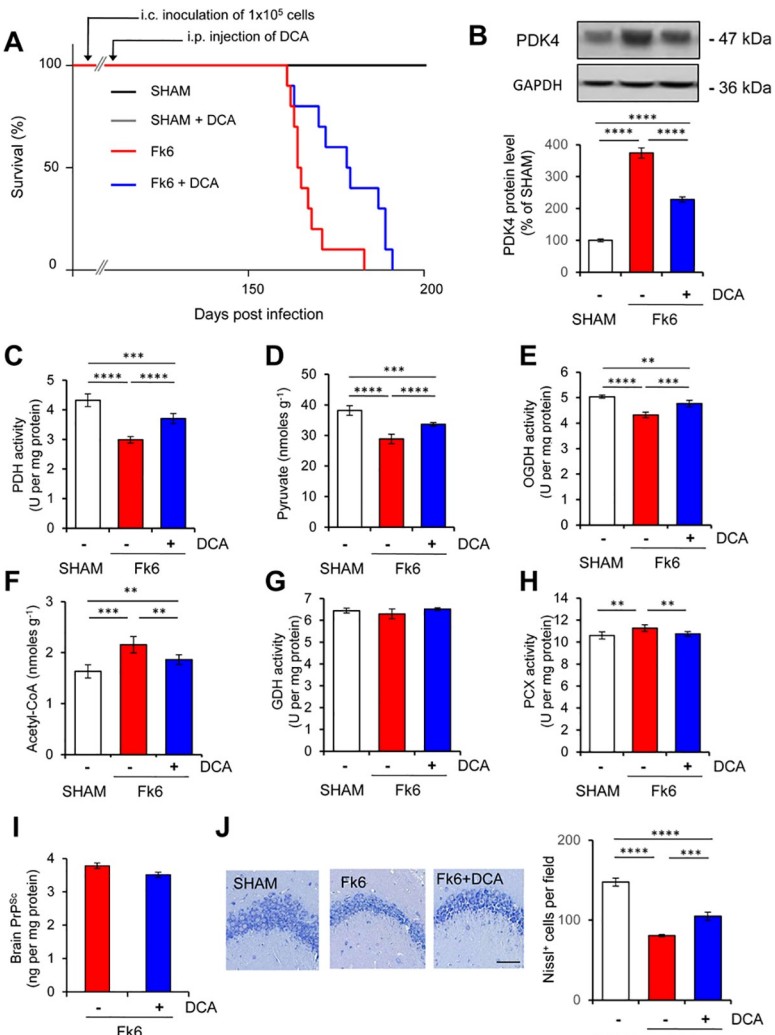

**Fig 6. PrP^Sc down-regulates PDH activity and glucose oxidative degradation in the hippocampus of prion-infected mice. (A)** Survival curves of mock- (SHAM) and prion-inoculated (Fk6) mice via the intracerebral route (i.c.) infused or not with the PDK4 inhibitor DCA by intraperitoneal injection (i.p.) starting 130 d after infection (100 mg per kg body weight per day; 0.25 μl h$^{-1}$, n = 10). **(B)** Representative western blot and quantification histogram of PDK4 protein expression level in the hippocampus of C57Bl6/J mice infected or not with Fk6 prions infused or not with DCA. GAPDH was used for normalization (n = 6). **(C to H)** PDH activity **(C)**, pyruvate amount **(D)**, OGDH activity **(E)**, acetyl-CoA level **(F)**, GDH activity **(G)**, and PCx activity **(H)** in the hippocampus of C57Bl6/J mice infected or not with Fk6 prions infused or not with DCA (n = 6). **(I)** Postmortem quantification of proteinase K-resistant PrP in brains of Fk6-mice infused or not with DCA (n = 10). **(J)** Nissl staining and quantification histogram of viable neurons in the hippocampus of C57Bl6/J mice infected or not with Fk6 prions infused or not with DCA at the end stage of the disease (n = 10). Scale bar = 100 μm. Data are the mean ± SEM. n.d. not detected. * denotes $p < 0.05$, ** $p < 0.01$ and **** $p < 0.0001$.

Finally, we showed that chronic intraperitoneal injection of the PDK4 inhibitor DCA (100 mg per kg body weight per day; 0.25 μl h$^{-1}$) starting 130 days after infection and before the onset of clinical signs (140 days) delayed mortality in Fk6-mice of 14 ± 3 days compared to untreated infected mice (178.5 ± 3.5 days *vs.* 164.5 ± 2.0 days, n = 10, $p < 0.05$, **Fig 6A**) with no overt sign of DCA toxicity. Upon PDK4 inhibition, the PDK4 protein level measured in the hippocampus at the end-stage of the disease decreased compared to untreated prion-infected mice (**Fig 6B**). Such a decrease of PDK4 likely reflects repression of PDK4 expression caused

by the reduction of fatty acids β-oxidation [53] in favor of glucose metabolism. Accordingly, PDK4 inhibition allowed PCx activity to return to the basal level measured in SHAM mice (**Fig 6H**). The enzymatic activities of PDH (**Fig 6C**) and OGHD (**Fig 6E**), as well as the concentrations of pyruvate (**Fig 6D**) and acetyl-CoA (**Fig 6F**), showed a tendency to normalize in the hippocampus of prion-infected mice infused with DCA. These overall data indicate the recovery of the glucose over FA metabolism flux balance in prion-infected mice treated with the PDK4 inhibitor. Of note, post-mortem quantifications of proteinase K-resistant PrP revealed no significant variation in PrP$^{res}$ level in the brain of Fk6-mice infused with DCA compared to untreated infected animals (**Fig 6I**). Despite no change in PrP$^{Sc}$ amount upon PDK4 inhibition, DCA treatment attenuated prion-induced neuronal loss in the hippocampus as assessed by increased Nissl staining of viable neurons compared to untreated infected mice (**Fig 6J**). This argues that PrP$^{Sc}$-induced deregulation of PDK4 activity and the subsequent metabolic abnormalities contribute to neurodegeneration in prion diseases. Further supporting those metabolic abnormalities caused by prion infection precede neuronal death, we measured in the hippocampus of Fk6-mice sacrificed at 130 days post-infection, *i.e.*, before DCA treatment starts, a rise in PDK4 protein level and a decrease in PDH activity compared to SHAM mice, but no significant neuronal loss (**S5 Fig**).

Altogether, these results indicate a loss-of-PrP$^{C}$ control of the metabolic orientation towards the oxidative degradation of glucose in favor of the fatty acids β-oxidation pathway in the brain of prion-infected mice. These data further show that the inhibition of PDK4 exerts beneficial effects in prion diseases.

## Discussion

Although corruption of normal functions of PrP$^{C}$ plays a central role not only in prion diseases but also in Alzheimer's disease, and possibly Parkinson's disease, much can still be learned about PrP$^{C}$ role(s). In this work, a global study that compared proteomes of 1C11 neuronal stem cells expressing or not PrP$^{C}$ reveals, for the first time to our knowledge that PrP$^{C}$ is a key factor in the regulation of two important metabolic pathways, the oxidative degradation of glucose and the β-oxidation of fatty acids, with consequent implications on oxidative stress.

Our proteomic approach highlights that the cellular energy metabolism is deeply affected by the absence of PrP$^{C}$ with increased expression (+20%) of glycolytic and fermentation enzymes contrasted by reduced expression of mitochondrial enzymes of the TCA cycle (-20%). For all enzymes tested, changes in their protein expression level in PrP$^{null}$-cells paralleled transcriptional variations. We expected that these modifications in the steady-state levels of glycolytic and mitochondrial enzymes would have induced a metabolic shift from complete oxidative degradation of glucose to incomplete degradation of glucose by lactate dehydrogenase (LDH) fermentation in PrP$^{null}$-1C11 cells. Such transition, observed even in the presence of oxygen and better known as the Warburg effect [65,66], is a cell adaptive state to compensate for the lack of energy normally produced by mitochondria. Unexpectedly, the NMR-based metabolomics approach and Seahorse experiments showed a reduced glycolytic flux in PrP$^{C}$-depleted cells. A higher level of glucose and a lower amount of lactate (*i.e.*, a lower acidification rate) were measured in cell culture media of PrP$^{null}$-1C11 cells *vs*. 1C11 cells (**S6 Fig**). In PrP$^{null}$-1C11 cells, impaired glycolytic flux is not associated with reduced expression of glucose transporter GLUT3. Instead, it depends on a deficit of PDH activity that originates from lower expressions of PDHA1 and DLD and a rise in the phosphorylation level of PDHA1 at Ser residues [51,52,59,67]. The decreased PDH activity and the subsequently reduced conversion rate of pyruvate into acetyl-CoA in PrP$^{null}$-cells lead to a lower basal OCR and the incapacity of mitochondria to respond to variations of external glucose concentration. Seahorse

experiments, however, indicated that the absence of PrP$^C$ had no impact on the intrinsic activity of the electron transporters of the respiratory chain, and did not affect the chemo-osmotic coupling of the respiratory chain to the ATP synthase.

Our data further reveal that regulation of PDH activity by PrP$^C$ required PrP$^C$-dependent upstream negative control of PDK4 expression, the main kinase in charge of PDHA1 phosphorylation [51,52,59]. In PrP$^C$-depleted cells, the expression level of PDK4 was indeed up-regulated at mRNA and protein levels (**S6 Fig**). Transcriptional PDK4 overexpression in the absence of PrP$^C$ would relate to the transcription factor PPARγ [53], whose expression was also strongly enhanced in PrP$^{null}$-cells (**S6 Fig**). We provide evidence that the concerted up-regulation of PPARγ and PDK4 was caused by loss of PrP$^C$ coupling to the cAMP/PKA signaling pathway [57] (**S6 Fig**), because PKA stimulation with dbcAMP in PrP$^{null}$-cells reduced both their mRNA and protein levels. We previously showed that PrP$^C$ signaling regulates the activity of the transcription factor cAMP Responsive Element Binding (CREB) [68]. As CREB was reported to reduce PPARγ expression in the liver [69] via HES-1 [54], PrP$^C$ may thus favor the preferential use of glucose for energetic purposes by toning down the expression of PPARγ and thereby that of PDK4 through PrP$^C$ positive control of the cAMP/PKA/CREB signaling cascade.

Moreover, PDK4 overexpression could be maintained by the fatty acids β-oxidation pathway [53], whose intensity is ~20% increased in PrP$^{null}$-1C11 cells (**S6 Fig**). The imbalance between oxidative degradation of glucose and the use of fatty acids would likely account for the excess of ATP (+65%) in PrP$^C$-depleted cells. Because ATP was shown to be a positive regulator of the PDK4 activity (for review, see [59] and references therein), the excess of ATP in PrP$^{null}$-cells would additionally amplify fatty acids use at the expense of glucose oxidative degradation.

The metabolic conversion of PrP$^{null}$-cells towards the preferential degradation of fatty acids is accompanied by the onset of oxidative stress conditions (**S6 Fig**). PrP$^{null}$-cells displayed an excessive ROS level (+50%) combined with reduced amount of major antioxidant systems, such as GSH. The rise of ROS in PrP$^C$-depleted cells may originate from mitochondria activity upon fatty acids degradation. Reduced PDH activity in the absence of PrP$^C$ might also contribute to the saturation of the respiratory chain transporters with electrons and leakage of electrons onto $O_2$ at the root of increased ROS production [70]. However, by showing a metabolic preference of PrP$^C$-expressing cells for glucose use at the expense of fatty acids, our study reveals a new side of PrP$^C$ function in the control of the cell redox status. Indeed, PrP$^C$ exerts a double control of the redox equilibrium by (i) governing the production of ROS through NADPH oxidase [36] and (ii) steering the cell energetic metabolism towards the degradation of glucose. These two PrP$^C$ functions fit within the more global cytoprotective role of PrP$^C$ [9,39,71].

Mitochondria dysfunction and metabolic abnormalities have been associated with many aggregate-prone protein neurodegenerative diseases (Alzheimer's disease (AD), Parkinson's disease (PD), prion diseases. . .) [15–20]. Another common hallmark of all these diseases is an imbalance of the redox equilibrium and the onset of oxidative stress conditions [72]. Several organelles, such as mitochondria or peroxisomes, and/or enzymatic systems, including the NADPH oxidase, the superoxide dismutase, or the monoamine oxidases, have already been implicated for the increased neuronal redox status in the mentioned diseases [15,73,74]. It has to be noted that PrP$^C$ can bind PrP$^{Sc}$, but also Aβ, and pathological α-synuclein, and the corruption of PrP$^C$ signaling function via these abnormally folded/aggregated proteins lies at the root of neurodegeneration [2–4,7,62,63,75]. In this context, the alteration of PrP$^C$'s metabolic function by PrP$^{Sc}$ or other amyloids may interfere with the fine-tuning of the equilibrium between the use of carbohydrates *vs*. fatty acids, and it may be involved in neurodegenerative

processes. According to our present results on prion-infected mice, PrP$^{Sc}$ affects PDK4 and the mitochondrial PDH complex. The reduced enzymatic activity of PDH leads to reduced oxidative degradation of glucose and increased fatty acids β-oxidation in the hippocampus of prion-infected mice. Thus, it is possible that the metabolic reprogramming of prion-infected neurons towards fatty acids β-oxidation would generate high levels of reactive oxygen species, as well as end-products of the β-oxidation pathway [60], and all of this may contribute to neurodegeneration. Importantly, the inhibition of PDK4 with DCA extends survival of prion-infected mice. This posits PDK4 as a potential therapeutic target to fight prion diseases and possibly other amyloid-based neurodegenerative diseases by rescuing a normal glucose oxidative flux and limiting the use of fatty acids.

For the first time, our study provides evidence that non-pathological cellular prion protein PrP$^{C}$ is a regulator of glucose metabolism. PrP$^{C}$ orientates in cells energetic metabolism towards mitochondria oxidative degradation of glucose. Such PrP$^{C}$ action depends on PrP$^{C}$ coupling to the cAMP/PKA signaling pathway that attenuates PDK4 expression and activity and thereby optimizes downstream PDH activity for efficient transfer and conversion of pyruvate into acetyl-CoA in mitochondria. PrP$^{C}$-driven preferential use of glucose in cells limits fatty acids β-oxidation and the onset of oxidative stress conditions. Through its positive action on glucose metabolism, PrP$^{C}$ thus exerts an antioxidant role. According to our results on prion infection, pathogenic prions PrP$^{Sc}$ provoke a PrP$^{C}$ loss-of-function at the root of the metabolic abnormalities recorded in the brain of prion-infected mice. Metabolic reprogramming of prion-infected brain towards fatty acids consumption contributes to neurodegeneration. Of note, pharmacological inhibition of PDK4 in mice was performed at a late stage of prion disease, *i.e.*, day 130 post-infection, that is, 10 days before symptoms start and ~30 days before the death of mice. In the absence of any early diagnosis of Creutzfeldt-Jakob disease in humans, the rationale of such a late stage treatment approach in mice was to get as close as possible to the therapeutic time window for patients who just declared clinical signs of Creutzfeldt-Jakob disease. In those conditions, DCA infusion was sufficient to counteract the metabolic abnormalities, protect neurons from neurodegeneration, and prolong the lifetime of prion-infected mice. Face today's absence of any medicine to combat prion diseases, our data thus suggest dichloroacetate, a drug already tested in diverse clinical trials for the treatment of congenital lactic acidosis, and several solid cancers [76,77], would be used as a symptom-modifying drug for treating Creutzfeldt-Jakob disease patients.

## Materials and methods

### Ethics statement

Adult wild type FVB and PrP$^{0/0}$ mice were bred and underwent experiments, respecting European guidelines for the care and ethical use of laboratory animals (Directive 2010/63/EU of the European Parliament and of the Council of 22 September 2010 on the protection of animals used for scientific purposes). Adult C57Bl/6J mice were bred and underwent experiments in level-3 biological risk containment, respecting European guidelines for the care and ethical use of laboratory animals (Directive 2010/63/EU of the European Parliament and of the Council of 22 September 2010 on the protection of animals used for scientific purposes). Six 8-week-old male C57Bl/6J mice per group were inoculated intracerebrally with 20 μl of sample containing cell extracts (1 x 10$^5$ 1C11 cells infected or not with Fukuoka-1 prion strain) [63]. Cells were submitted to three freeze-thaw cycles and suspensions were sonicated for 2 min (Cuphorn sonicator; Nanolab Inc, Waltham, MA, USA). All animal procedures were approved by the Animal Care and Use Committee at Basel University (Switzerland, #5787–2016062207437674).

## Chemicals

The PDK4 inhibitor sodium dichloroacetate (DCA) and the PKA activator dibutyryl cyclic-AMP (dbcAMP) were from Sigma-Aldrich/Merck, USA.

## Culture of 1C11 and PC12 cells and PrP[C] silencing

1C11 cells and PrP[null]-1C11 cells [31] were grown in Dulbecco's modified Eagle's medium (DMEM) containing 5.5 or 25 mM glucose and supplemented with Glutamax 1X and 10% fetal calf serum (FCS, Biochrom, UK) at 37˚C in a humid atmosphere with 10% $CO_2$. In PrP[null]-1C11 cells, PrP[C] is chronically repressed by at least 95% compared to 1C11 parental cells (**S7 Fig**) due to the constitutive expression of an shRNA targeting PrP mRNA [31]. PC12 cells were induced to differentiate with NGF (50 ng ml[-1]) for 5 days and transfected with PrP[C]-siRNA (siPrP) or Scrambled siRNA (siScr) [31,78].

## Proteomic analysis by quantitative mass spectrometry

1C11 and PrP[null]-1C11 cells were homogenized in SDS-containing lysis buffer (2% SDS, 62.5 mM HEPES/NaOH, pH 8.0; preheated to 90˚C), with the use of 1.0 mm zirconia beads and a Mini-BeadBeater-8 (Biospec Products Inc., USA). Following three cycles of 1 min bead beading, the lysates were further incubated at 90˚C to deactivate residual enzymatic activities in the extracts. Protein levels were adjusted by the bicinchoninic acid method (BCA) colorimetric assay (Thermo Scientific, Canada) before sample preparation for global proteome analyses. Protein precipitation, denaturation, reduction, alkylation, and digestion were performed as in [79]. MS grade trypsin was from Thermo Scientific. Tryptic peptides were covalently modified with the TMTsixplex isobaric label reagent set (Thermo Scientific) according to the protocol supplied by the manufacturer. The quantitative mass spectrometry was performed as in [48].

## Cluster analysis

Hierarchical clustering was conducted by the DAVID software–Database for Annotation, Visualization, and Integrated Discovery. More specifically, the KEGG pathway analysis was used to determine the cellular functions affected by PrP[C] depletion.

## RNA isolation and real-time quantitative RT-PCR analyses

Total RNA was isolated using an RNeasy mini kit according to the manufacturer's instructions (Qiagen, Germany). The first-strand cDNA synthesis was performed with the Prime Script RT Master Mix kit (Takara Bio Europe, France). Quantitative real-time PCR was performed at 60˚C using Takyon ROX SYBR MasterMix (Eurogentec, Belgium) in the CFX384 Touch Real-Time PCR Detection System (Bio-Rad, France). Primers used in RT-qPCR analyses are listed in **S2 Table.** Rplp0 was used as the internal control.

## Cell extract preparation and western blot analysis

Cells were washed in phosphate-buffered saline (PBS) 1X buffer (Invitrogen, ThermoFisher Scientific, USA) and incubated for 30 min at 4˚C in lysis buffer (50 mM Tris-HCl pH 7.4, 150 mM NaCl, 5 mM EDTA, 1% Triton X-100, and cocktails of protease and phosphatase inhibitors [Roche, Switzerland]). After centrifugation of the lysate (14,000 g, 30 min), the concentration of the proteins in the supernatant was measured with the BCA method (Pierce, Thermo Fisher Scientific, France). Twenty micrograms of proteins were resolved by 10% SDS/PAGE and transferred to nitrocellulose membranes (Bio-Rad). Membranes were blocked with 3% non-fat dry milk or 5% bovine serum albumin (depending on the antibody) in PBS 1X

containing 0.1% Tween 20 for 1 h at room temperature and then incubated overnight at 4°C with primary antibody (S3 Table). Bound antibodies were revealed by enhanced chemiluminescence (ECL, Bio-Rad) detection using a secondary antibody coupled to *horseradish peroxidase* (HRP, Southern Biotech, USA). To standardize the results, membranes were rehybridized with an anti-α-tubulin antibody (Proteintech, UK). Protein levels were quantified using ImageQuantTL software. For all analyses, the reference level (100%) was fixed using unexposed 1C11 cells.

## NMR analyses of metabolites present in the cell culture medium

Cells were grown for 3 days to ~80% confluence in DMEM-Glutamax-FCS containing 25 mM glucose. The cell culture medium was collected for the NMR analysis. Aliquots of medium were mixed with 0.005% TSP buffer (TSP 0,005%, $KH_2PO_4$, $Na_2HPO_4$, $D_2O$) before performing NMR spectroscopy [80]. A Multivariate analysis was performed using SIMCA software. A Supervised Orthogonal Projections to Latent Structures Discriminant Analysis (OPLS-DA) [49] was carried out, taking into account the class of the different samples to maximize class discrimination. The quality of the obtained models was assessed with the quality factors given by SIMCA software, *i.e.* the cumulative explained variance of observed data X, $R^2X(cum)$, the cumulative explained variance of assignment Y, $R^2Y(cum)$, and the estimate of predictive ability, $Q^2(cum)$. The goodness of the models was further tested using CV-ANOVA [81] with a p-value $< 0.05$ being considered statistically significant. In order to get insight into the metabolites identified to be important in the OPLS-DA model, univariate analysis was done using Student's t-tests and ANOVA as implemented in MetaboAnalyst [82,83].

## Seahorse analyses

Seahorse $XF^e96$ Extracellular Flux Analyzer (Seahorse Biosciences, Agilent Technology, USA) was used to measure the oxygen consumption rate (OCR) and the extracellular acidification rate (ECAR). To this end, 1C11 and $PrP^{null}$-1C11 cells (15 x $10^3$ cells per well) were grown in DMEM-Glutamax-FCS containing 5.5 or 25 mM glucose in a 96-well plate for one day. On the day of the assay, the culture medium was changed to XFi Assay Medium supplemented with 1 mM pyruvate, 1X Glutamax, 1% FCS and glucose at a concentration of 5.5 or 25 mM. Four baseline measurement cycles were followed by the sequential injection of oligomycin (10 mg $ml^{-1}$, Sigma-Aldrich), carbonyl cyanide 3-chlorophenylhydrazone (CCCP, 5 µM, Sigma-Aldrich), and antimycin A (10 µM, Sigma-Aldrich) with four measurement cycles between each injection and four final measurement cycles. OCR and ECAR were simultaneously recorded and calculated by the Seahorse $XF^e96$ Software, Wave (Seahorse Biosciences). The mean values are based on 6 replicates.

## Palmitate β-oxidation

Cells (5 to 8 x $10^3$ cells per well) grown in 24-well plates for four days were washed 3 times in PBS 1X and PBS 1X containing 100 µM of 9,10(n)-[$^3$H] palmitic acid (60 Ci/mmol, Perkin Elmer Life Science, USA) was added in each well. After 2 h of incubation at 37°C, the medium was collected and added to cold 10% trichloroacetic acid. The tubes were centrifuged for 10 min at 2,200 g at 4°C and supernatants were removed, then mixed with 6 N NaOH, and applied to the ion-exchange resin. The columns were washed twice with water and the radioactivity of eluates was counted. Protein quantity was determined by the Lowry method to standardize the results.

## Measurement of ATP cell content

ATP content in 1C11 precursors cells and PrP$^{null}$-1C11 cells ($2.5 \times 10^5$ cells) grown for 4 days in DMEM-Glutamax-FCS containing 25 mM glucose was assessed using the Enliten ATP Assay System Bioluminescence Detection Kit according to the manufacturer's instructions (Promega, USA). The luminescence was recorded at $\lambda_{em}$ = 560 nm (integrate period = 10 sec, slit width = 5 nm) using a Cary Eclipse (Varian Inc., Agilent Technology).

## Measurement of cellular ROS level

Production of ROS in 1C11 precursor cells and PrP$^{null}$-1C11 cells was assessed using the intracellular fluorogenic reagent CM-H$_2$DCFDA according to the manufacturer's instructions (Molecular Probes, Thermo Fisher Scientific). The fluorescence was recorded in cell lysates at $\lambda_{em}$ = 528 nm (slit width = 5 nm) after excitation at $\lambda_{exc}$ = 507 nm (slit width = 5 nm) using a Cary Eclipse.

## Measurement of intracellular reduced glutathione

The level of GSH was determined using the GSH sensitive probe Celltracker Green CMFDA (Molecular Probes, Thermo Fisher Scientific). Cells were washed twice with PBS 1X and incubated for 30 min at 37°C in PBS 1X in the presence of 1 μM fluorogenic reagent. PBS was removed and the cells were left to reconstitute in DMEM, Glutamax 1X and 10% FCS for 30 min at 37°C before lysis. Fluorescence intensity of cell lysates was recorded at $\lambda_{em}$ = 517 nm (slit width = 5 nm) after excitation at $\lambda_{exc}$ = 492 nm (slit width = 5 nm) using a Cary Eclipse.

## Isolation and culture of hippocampal neurons

Cultures of adult hippocampal neurons were established from 3-month-old FVB and PrP$^{0/0}$-FVB mice as previously reported [84]. Neurons were left to regenerate for ten days in culture before DCA treatment.

## Chronic intraperitoneal injection of DCA into mice

Mice were fasted overnight but allowed water ad libitum before the experiment. They were then anesthetized with isoflurane inhalation, and a midline incision was performed to insert into the peritoneum the polyethylene catheter of an osmotic pump (Alzet, Cupertino, CA, USA). DCA (100 mg kg$^{-1}$ per day) or vehicle (sterile DMEM supplemented with 25 mM PIPES and HEPES) was administered at a flow rate of 0.25 μL h$^{-1}$. Pumps were replaced every 3 weeks.

## Hippocampus isolation from C57Bl/6J mice and extract preparation

Mice were decapitated under light isoflurane anesthesia. Brains were removed and hippocampus dissected out and immediately frozen in liquid nitrogen to efficiently preserve tissues and metabolites [85]. The cytosolic fraction was extracted into a 3.6% perchloric acid solution and neutralized with KHCO$_3$ (3 M) [86]. Mitochondria were isolated from the cytosolic fraction [87]. The mitochondrial fraction was resuspended in 1 ml mitochondrial isolation buffer. Both cytosolic and mitochondrial fractions were stored at -80°C until their use for the measurement of metabolic enzymatic activities.

## Nissl staining of viable neurons in mouse hippocampus

Mice were perfused with 10% formalin. The brain tissues were fixed in 10% formalin overnight at 4˚C and subsequently embedded in paraffin. Hippocampus slices were made (4 μm thick) and paraffin sections were deparaffinized, rehydrated, and stained with 5% Cresyl Violet acetate followed by wash and imaging with an optical microscope. Cells positive for cytoplasmic Nissl staining with loose chromatin and prominent nucleoli were considered healthy neurons. The data represent the number of cells per field.

## Measurement of PDH, 2-OGDH, GDH, and PCx activities

The activities of all enzymes were measured with the SpectraMax 190 Microplate reader (Molecular Devices) via continuous spectrophotometric assays. All enzyme activities were normalized to protein content, measured via a Pierce bicinchoninic acid (BCA) assay (Thermo-Fisher Scientific). PDH (EC 1.2.4.1) activity was measured using the 3-(4,5-dimethyl-2-thiazolyl)-2,5-diphenyl-2H tetrazolium bromide (MTT) and phenazine methosulfate method [88]. The activity of 2-oxoglutarate dehydrogenase (2-OGDH) was measured via the reduction of nicotinamide adenine dinucleotide (NAD) in 75 mM Tris HCl (pH 8), 1 mM ethylenediaminetetraacetic acid, 0.5 mM thiamine pyrophosphate, 1.5 mM coenzyme A, 4 mM NAD, 1 mM DTT, and 2 mM calcium chloride, and initiated with 15 mM 2-OG. Glutamate dehydrogenase (GDH) activity was measured through the oxidation of reduced β-nicotinamide adenine dinucleotide (β-NADH). Pyruvate carboxylase (PCx) activity was measured through the production of $TNB^{2+}$ at a wavelength of 412 nm. The reaction mix contained 50 mM Tris HCl (pH 8), 50 mM sodium bicarbonate, 5 mM $MgCl_2$, 5 mM sodium pyruvate, 5 mM ATP, 0.5 mM 5,5'-dithiobis-(2-nitrobenzoicacid), and 5 U/mL citrate synthase (Sigma-Aldrich). The reaction was initiated with 0.1 mM acetyl-CoA.

## Measurement of pyruvate amount

Twenty microliters of the cytosolic fraction were taken for determination of pyruvate and a commensurate amount of the $^{13}C$-labeled internal standard (2–3 fold over endogenous metabolite) was added for quantitative analysis by gas chromatography-mass spectrometry. The aqueous solution of the samples was reduced to dryness under a stream of nitrogen (99.99%) and reacted with tertiary butyl dimethylsilyl chloride to form the silyl ester [85]. One microliter of the derivatized sample was injected onto a 30 m capillary gas chromatography column and analyzed in the electron impact mode on a quadrupole mass spectrometer. Individual metabolites were quantified using the ratio of the area counts of the most prominent ion fragments of the unlabeled compound referenced to the analogous ion of the labeled internal standard.

## Measurement of acetyl-CoA amount

A PicoProbe acetyl-CoA fluorometric assay kit (BioVision) and a SpectraMax Gemini EM Microplate Reader (Molecular Devices) were used to measure acetyl-CoA amount.

## PrP<sup>res</sup> quantification

The amount of proteinase K–resistant PrP (PrP<sup>res</sup>) in brain extracts of Fk6-infected mice infused or not with DCA were determined using a PrP-specific sandwich ELISA [3,62,63] after proteinase K digestion (10 μg ml$^{-1}$) for 1 h at 37˚C.

### Statistical analyses

Statistical analysis (t-test for 2 groups analysis and 1-way ANOVA for 3 or more groups) to determine significance was performed using GraphPad Prism software (GraphPad Software Inc., USA). Values are given as means ± S.E.M. and a p-value < 0.05% was considered significant. Survival times were analyzed by Kaplan-Meier survival analysis using a log-rank test for curve comparisons.

## Supporting information

**S1 Fig.** (**A**) Cellular functions and (**B**) metabolic pathways mainly affected by the absence of PrP$^C$ in 1C11 neuronal stem cells. Data are presented as the percentage of proteins involved in a specific cell function or metabolic pathway, whose expression is affected by the silencing of PrP$^C$.
(TIF)

**S2 Fig.** Representative Western blots of AldoA (**A**), TPI (**B**), Eno3 (**C**), PGAM1 (**D**), ADH (**E**), OGDH (**F**), CS (**G**), PDHA1 (**H**), and DLD (**I**) expression in 1C11 and PrP$^{null}$-1C11 cells. α-tubulin was used for normalization.
(TIF)

**S3 Fig.** Relative mRNA levels of PDK1, PDK2, PDK3 and PDK4 between 1C11 and PrP$^{null}$-1C11 cells as assessed by RT-qPCR (n = 3).
(TIF)

**S4 Fig.** (**A**) Representative western blot and quantification histogram of PDK4 protein expression level and (**B**) PDH activity in PC12 neuronal cells transiently silenced for PrP$^C$ expression (SiPrP) or not (SiScr) (n = 6). PC12 cells were treated with DCA (2 mM) for 6h. (**C**) Representative western blot and quantification histogram of PDK4 protein expression level and (**D**) PDH activity in primary cultures of hippocampal neurons isolated from adult FVB and PrP$^{0/0}$-FVB mice and left to regenerate for 10 days (n = 6). Hippocampal neurons were treated with DCA (2 mM) for 6 h. GAPDH was used for normalization in western-blot experiments. Data are the mean ± SEM. *** denotes p < 0.001 and **** p < 0.0001.
(TIF)

**S5 Fig.** (**A**) Representative western blot of PDK4 protein expression level, (**B**) PDH activity, and (**C**) Nissl staining of viable neurons in the hippocampus of C57Bl6/J mice infected or not with Fk6 prions sacrificed at 130 days post-infection (n = 4). Scale bar = 100 μm. GAPDH was used for normalization in western-blot experiments. Data are the mean ± SEM. **** denotes p < 0.0001.
(TIF)

**S6 Fig. Scheme representation of metabolic abnormalities in the absence of PrP$^C$.** In PrP$^{null}$-cells, loss of PrP$^C$ coupling to the cAMP/PKA signaling pathway abrogates PrP$^C$ negative control of PPARγ and PDK4 expression. The subsequent rise in PDK4 level and activity leads to a reduction of PDH complex activity associated with a rise of phosphorylation of the PDHA1 subunit. This provokes a decrease of the glycolytic flux in favor of an increase in the fatty acids β-oxidation rate, which combined to the excess of synthesized ATP sustain high PDK4 activity. The in fuel preference of PrP$^{null}$-cells towards the use of fatty acids is accompanied by the onset of oxidative stress conditions. In PrP$^C$-expressing cells, PrP$^C$ coupling to cAMP/PKA signaling tones down PPARγ/PDK4 expressions, which equilibrates carbohydrate and fatty acid degradations, and thereby confers anti-oxidative stress function to PrP$^C$.
(TIF)

**S7 Fig.** Representative western blot and quantification histogram showing PrP$^C$ depletion (>95%) in 1C11 cells chronically silenced for PrP$^C$ upon constitutive expression of a siRNA targeting PrP mRNA compared to parental 1C11 cells [31] using Sha31 PrP antibody (n = 3). α-tubulin was used for normalization. Data are the mean ± SEM. **** denotes p < 0.0001. (TIF)

**S1 Table.** Relative steady-state abundance levels of proteins expressed differentially between PrP$^{null}$- and 1C11 cells (ratio < 0.84 and > 1.25). (XLSX)

**S2 Table.** RT-qPCR primers. (XLSX)

**S3 Table.** Antibodies. (XLSX)

## Acknowledgments

We deeply thank F. d'Agostini for prion infection of mice and isolation of hippocampus, as well as R. Kettler and G. Zurcher for all metabolic data acquisition and analyses with prion-infected mice. We thank Pr. S. Blanquet and Dr. D. Tampellini for critical reading of the manuscript and helpful discussions. mRNA and protein studies were performed at the Cyto2BM core facility of BioMedTech Facilities INSERM US36 / CNRS UMS2009 / Université de Paris.

## Author Contributions

**Conceptualization:** Hélène Arnould, Anne Baudry, Mathéa Pietri, Jean-Marie Launay, Benoit Schneider.

**Data curation:** Hélène Arnould.

**Formal analysis:** Hélène Arnould, Vincent Baudouin, Anne Baudry, Luiz W. Ribeiro, Mathéa Pietri, Gildas Bertho, Jean-Marie Launay, Gerold Schmitt-Ulms, Benoit Schneider.

**Funding acquisition:** Benoit Schneider.

**Investigation:** Hélène Arnould, Vincent Baudouin, Hector Ardila-Osorio, Cédric Caradeuc, Cynthia Soultawi, Declan Williams, Marjorie Alvarez, Fatima Djouadi, Mireille Laforge, Gildas Bertho, Jean-Marie Launay, Gerold Schmitt-Ulms.

**Methodology:** Hélène Arnould, Hector Ardila-Osorio, Cédric Caradeuc, Cynthia Soultawi, Declan Williams, Carole Crozet, Fatima Djouadi, Mireille Laforge, Gildas Bertho, Jean-Marie Launay, Gerold Schmitt-Ulms.

**Project administration:** Benoit Schneider.

**Resources:** Carole Crozet, Jean-Marie Launay.

**Supervision:** Anne Baudry, Benoit Schneider.

**Validation:** Hélène Arnould, Mireille Laforge, Gildas Bertho, Jean-Marie Launay, Gerold Schmitt-Ulms, Benoit Schneider.

**Visualization:** Hélène Arnould, Vincent Baudouin, Luiz W. Ribeiro, Mathéa Pietri.

**Writing – original draft:** Anne Baudry, Odile Kellermann, Benoit Schneider.

**Writing – review & editing:** Anne Baudry, Mathéa Pietri, Odile Kellermann, Benoit
Schneider.

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
