## [Decision Letter · Decision Letter 0]

5 Aug 2021

Dear Dr. Schneider,

Thank you very much for submitting your manuscript "Loss of prion protein control of glucose metabolism promotes neurodegeneration in model of prion diseases." for consideration at PLOS Pathogens. As with all papers reviewed by the journal, your manuscript was reviewed by members of the editorial board and by several independent reviewers. In light of the reviews (below this email), we would like to invite the resubmission of a significantly-revised version that takes into account the reviewers' comments.

In particular, while the reviewers #1 and #2 raise only minor or no concerns, the reviewer #3 raises a number of serious points. I ask you to respond, carefully and point by point, to the criticisms and to consider the option of using primary cells from knock-out mice to corroborate your findings in a third-cell model (or at least to discuss their use). The subject of the study is of great interest and the results deserve to be published, therefore, a very open discussion of the limitations inherent to the experimental approach would help readers to correctly interpret the results.

We cannot make any decision about publication until we have seen the revised manuscript and your response to the reviewers' comments. Your revised manuscript is also likely to be sent to reviewers for further evaluation.

Sincerely,

Umberto Agrimi

Associate Editor

PLOS Pathogens

Neil Mabbott

Section Editor

PLOS Pathogens

Kasturi Haldar

Editor-in-Chief

PLOS Pathogens

orcid.org/0000-0001-5065-158X

Michael Malim

Editor-in-Chief

PLOS Pathogens

orcid.org/0000-0002-7699-2064

Reviewer's Responses to Questions

**Part I - Summary**

Reviewer #1: Using a proteomic approach the authors describe that PrPC may contribute to the regulation of the energetic metabolism by leaning cells towards mitochondrial oxidative degradation of glucose. As a result, prion-induced activation of pyruvate dehydrogenase kinase 4 (PDK4) and downstream reduction in mitochondria pyruvate dehydrogenase activity may promote a metabolic shift from glucose oxidative degradation to pro-oxidant fatty acids β-oxidation contributing to prion pathogenesis.

Reviewer #2: This is an interesting manuscript reporting a PrPC-mediated, PDK4-dependent imbalance between glucose oxidative degradation and fatty acid β-oxidation as a consequence of prion infection. The study employs different experimental paradigms, two different cell models, and prion-infected mice. It provides evidence that the pharmacological inhibition of PDK4 with dichloroacetate (DCA) slightly extended the survival of prion-infected mice.

The manuscript is well written, and, overall, the conclusions appear consistent with the results.

It is also evident that the authors addressed all the previous concerns raised by the reviewers.

Reviewer #3: The authors present a novel function for PrPC: the regulation of energy metabolism. They show that PDK4/PDH activities are altered by PrP loss in the 1C11 cell line, and they now present some corroborating evidence from PC12 cells. They show that PDK4 is also overexpressed in prion infected mice and that the levels can be returned to normal, and the lifespan of the mice extended, by pharmacological intervention using a PDK4 inhibitor. The subject of PrPC function, and its role in prion diseases, are matters of interest to the field. The extension of lifespan by more than two weeks by initiating drug treatment so late in the disease process is impressive. However, there remain several major weaknesses, which are still not addressed in the revised manuscript.

**Part II – Major Issues: Key Experiments Required for Acceptance**

Reviewer #1: The authors have revised the manuscript according to reviewers' suggestion and now it is very much improved.

The authors have satisfactorily answered all my comments and suggestions.

Reviewer #2: None

Reviewer #3: 1- The authors do not reconcile the findings in 1C11 cells with transcriptomic data that are available from Zurich-3 PrP KO mice. These data are reported in Nuvolone et al. (2016), which is referenced within the manuscript. Examination of these data shows that the relevant genes not dysregulated in the Zurich-3 mice. This discrepancy seems to undermine the conclusions of the paper. The authors suggest in the Introduction the general argument that embryonic compensatory mechanisms may play a role in the lack of phenotype of PrP KO mice, but they do undertake a detailed discussion of those transcripts that are altered in the Zurich 3 mice.

2- The use of shRNA (1C11 cells) or siRNA (PC12 cells) are not the most up-to-date or effective ways to eliminate gene expression. The authors should use CRISPR-Cas to completely ablate expression of PrP (either via introduction of indels or by excision of the gene). The metabolic changes should then be reversible by re-introduction of PrP expression. In addition, the concern about clonal variability of the cell lines (raised by a previous reviewer) is not adequately addressed. Both 1C11 and PC12 are transformed cells lines with abnormal and unstable ploidy, so clonal variability is a serious limitation. The authors would be better off using primary neurons from PrP KO mice.

3- As pointed out by the previous reviewers, the data from prion-infected mice may reflect non-specific metabolic alterations secondary to prion infection, rather a specific loss of PrP function. This concern is still not adequately addressed. In this regard, the title (which does not appear to have been changed) and overall conclusion that “Loss of prion protein control of glucose metabolism promotes neurodegeneration in a model of prion diseases” in unjustified. A long-standing question in the field has been whether prion diseases are due to a loss-of-function of PrPC or a gain-of-function of PrPSc. Animals devoid of PrP display, at most, minimal phenotypic abnormalities, and certainly not any features of a prion disease. In addition, there is evidence that significant levels of PrPC are present even at end stage of disease (Mays et al., 2015 doi:10.1128/JVI.02142-15). These findings argue strongly against loss of PrPC function a major contribution to the phenotype of prion diseases. This conclusion, which is diametrically opposed to the title of the paper, remains a fundamental problem with this study.

**Part III – Minor Issues: Editorial and Data Presentation Modifications**

Reviewer #1: N/A

Reviewer #2: My only suggestion would be to tone down the final outcome of the study. In fact, to my knowledge, DCA is not approved by the FDA as a cancer treatment, and its current use is exclusively for congenital lactic acidosis. More generally, the therapeutic benefits of DCA in lactic acidosis, cancer, and post-ischemic heart recovery have been quite controversial. I recommend the authors to explain in a more critical perspective the known limits and the possible side effects (e.g., peripheral neuropathy) of DCA in the clinical practice, also in light of the limited effects reported in this study.

Reviewer #3: The paper is poorly written and often contains confusing statements.

PLOS authors have the option to publish the peer review history of their article (what does this mean?). If published, this will include your full peer review and any attached files.

Reviewer #1: No

Reviewer #2: No

Reviewer #3: No
---

## [Editor Report · Decision Letter 1]

29 Sep 2021

Dear Dr. Schneider,

We are pleased to inform you that your manuscript 'Loss of prion protein control of glucose metabolism promotes neurodegeneration in model of prion diseases.' has been provisionally accepted for publication in PLOS Pathogens.

Best regards,

Umberto Agrimi

Associate Editor

PLOS Pathogens

Neil Mabbott

Section Editor

PLOS Pathogens

Kasturi Haldar

Editor-in-Chief

PLOS Pathogens

orcid.org/0000-0001-5065-158X

Michael Malim

Editor-in-Chief

PLOS Pathogens

orcid.org/0000-0002-7699-2064
---

## [Editor Report · Acceptance letter]

30 Sep 2021

Dear Dr. Schneider,

We are delighted to inform you that your manuscript, "Loss of prion protein control of glucose metabolism promotes neurodegeneration in model of prion diseases.," has been formally accepted for publication in PLOS Pathogens.

Best regards,

Kasturi Haldar

Editor-in-Chief

PLOS Pathogens

orcid.org/0000-0001-5065-158X

Michael Malim

Editor-in-Chief

PLOS Pathogens

orcid.org/0000-0002-7699-2064